# Maternal Health Care Service Utilization in the Post-Conflict Democratic Republic of Congo: An Analysis of Health Inequalities over Time

**DOI:** 10.3390/healthcare11212871

**Published:** 2023-10-31

**Authors:** Dieudonne Bwirire, Inez Roosen, Nanne de Vries, Rianne Letschert, Edmond Ntabe Namegabe, Rik Crutzen

**Affiliations:** 1Department of Health Promotion, CAPHRI Care and Public Health Research Institute, Faculty of Health, Medicine and Life Sciences, Maastricht University, 6229 HA Maastricht, The Netherlands; inez.roosen@maastrichtuniversity.nl (I.R.); nanne.devries@maastrichtuniversity.nl (N.d.V.); rik.crutzen@maastrichtuniversity.nl (R.C.); 2Maastricht University, 6200 MD Maastricht, The Netherlands; r.letschert@maastrichtuniversity.nl; 3Faculté de Santé et Développement Communautaires, Université Libre des Pays des Grands Lacs (ULPGL), Goma 368, Democratic Republic of the Congo; ntabenamegabe2006@gmail.com

**Keywords:** maternal health care service utilization, trends, health inequalities, inequality measurement, post-conflict, Democratic Republic of Congo (DRC)

## Abstract

This study assessed inequality in maternal healthcare service utilization in the Democratic Republic of the Congo, using the Demographic and Health Surveys of 2007 and 2013–2014. We assessed the magnitude of inequality using logistical regressions, analyzed the distribution of inequality using the Gini coefficient and the Lorenz curve, and used the Wagstaff method to assess inequality trends. Women were less likely to have their first antenatal care visit within the first trimester and to attend more antenatal care visits when living in eastern Congo. Women in rural areas were less likely to deliver by cesarean section and to receive postnatal care. Women with middle, richer, and richest wealth indexes were more likely to complete more antenatal care visits, to deliver by cesarean section, and to receive postnatal care. Over time, inequality in utilization decreased for antenatal and postnatal care but increased for delivery by cesarean sections, suggesting that innovative strategies are needed to improve utilization among poorer, rural, and underserved women.

## 1. Introduction

The Alma-Ata Declaration of the World Health Organization (WHO) states that the existing gross inequalities in the health status of people are unacceptable and are, therefore, of common concern to all countries [1]. This establishes a standard of public commitment to make quality health care accessible for all [2]. The Alma-Ata Declaration was the forerunner of the Global Strategy for Health for All and the Sustainable Development Goals (SDGs). The SDGs are a collection of 17 interdependent global goals set by the United Nations (UN) General Assembly in 2015 for the year 2030. Specifically, SDG 3 is devoted to health, and one of its targets is to reduce the global maternal mortality rate to less than 70 per 100,000 live births by 2030 [3].

Improving maternal health is critical to fulfilling the aspiration to reach SDG 3 [4,5]. Significant progress is being made in improving millions of people’s health [6], and some improvements have also been observed in maternal health. However, despite this progress, particularly in sub-Saharan Africa (SSA), the number of maternal deaths explained by a lack of access to and utilization of maternal health care services (MHCS) before, during, or after delivery [7] and socioeconomic inequality in health care use [8,9,10] are still high. The odds that a woman in SSA will die from complications related to pregnancy and childbirth is 1 in 20—an enormous difference from 1 in 6250 in the developed world [11]. Achievement of the 2030 SDGs is likely to be compromised if inequalities in health are not adequately addressed [12,13].

Countries that have experienced armed conflict often have the worst indicators of maternal mortality and very high levels of psychological impairment [14] and struggle to cope with the burden of diseases [15]. Often, there are significant health concerns, especially in maternal health care in these countries [16]. According to the United Nations (UN), efforts to improve maternal health are hindered by the presence of conflict, indicating that violence and instability can threaten governmental and international aid, further deterring health promotion [17]. While long-running conflicts have begun to decline or at least plateau, the underlying causes of many of these conflicts have not been addressed, and the potential for violence to flare up remains very real [18]. This can be observed in the Democratic Republic of the Congo (DRC), where many regions have known a series of destabilizing conflicts and wars [19]. This fragile environment may reinforce the existing cross-country maternal health inequities, particularly in the densely populated eastern regions [20]. According to the most recent Demographic and Health Survey (DHS 2013/2014), more than 200 ethnic groups live in the DRC, with the Bantu people as a large majority. The official language is French, and the capital city, Kinshasa, is the second-largest French-speaking city in the world. Four additional national languages are recognized: Kikongo, Lingala, Swahili, and Tshiluba. The majority of the country is Christian, mainly Catholic and Protestant. Despite a wealth of mineral resources, the DRC struggles with many socioeconomic problems, including high infant and maternal mortality rates, malnutrition, poor vaccination coverage, and lack of access to improved water sources and sanitation. Fertility remains high at more than five children per woman and is likely to remain high because of the low use of contraception and the cultural preference for larger families. Ongoing conflict, mismanagement of resources, and a lack of investment have resulted in food insecurity; almost 25% of children under the age of 5 were malnourished as of 2018. The overall coverage of basic public services—education, health, sanitation, and potable water—is very limited, with substantial regional and rural/urban disparities.

A limited number of studies have used all available Demographic and Health Surveys carried out in the DRC (EDS-RDC) datasets to examine the relationship between conflict and maternal healthcare service utilization in the DRC. We found one study by Ziegler et al. [21] that employed data from the 2007 and 2013–2014 EDS-RDC for this purpose. Of particular interest is that this study analyzed how predisposing, enabling, and need-based factors impact women’s antenatal care (ANC) and skilled birth attendant (SBA) usage, drawing theoretical insights from Andersen’s Behavioural Model of Health Care Utilization [22,23]. The study found that women in regions with extremely high levels of conflict were less likely to meet the WHO’s ANC recommendations compared to those in regions with moderate levels of conflict, suggesting that conflict-affected countries require context-specific interventions if progress is to be made toward achieving SDG 3.1. In the present study, we will focus on the inequality trends in the utilization of ANC, delivery services, and postnatal care (PNC), presenting a cross-sectional perspective.

To review progress concerning inequality in the utilization of MHCS and expand the evidence base to understand the problem better, some countries have conducted studies such as the Demographic and Health Surveys (DHS) and Household Income Expenditure Surveys (HIES) [24,25]. Others have monitored health inequalities between regions [26]. Specifically for the DRC, where disparities in MHCS exist between different provinces [17], a comprehensive overview of the MHCS utilization in those populations that are completely left behind is imperative. Therefore, we intend to make a theoretical contribution to the literature on health inequality that would also be useful to scholars beyond the Democratic Republic of Congo. For this paper, health inequalities refer to differences in the distribution of a specific factor (such as health status, income, and opportunities) between different population groups, while health inequities, on the other hand, are inequalities in which the outcome is unnecessary and avoidable, as well as unjust and unfair [27].

### Hypotheses and Aims of the Study

This study aims to assess health inequality trends in selected MHCS utilization variables in post-conflict DRC using publicly available DHS. Specifically, we address the following research questions (RQ):

RQ1. What is the magnitude of inequality in the utilization of MHCS in post-conflict DRC?

RQ2. What is the regional distribution of inequality in the utilization of MHCS in post-conflict DRC?

RQ3. What are the trends of inequalities in the utilization of MHCS in post-conflict DRC?

From these research questions, we further develop the following research hypotheses (H):

**Hypothesis H1.** 
*Health inequality in MHCS utilization has deteriorated in the DRC between 2007 and 2013–2014.*


**Hypothesis H2.** 
*There is an unequal distribution of MHCS utilization at the national, regional, or local level in the DRC. We assume that the western part of the country has better maternal health outcomes than the eastern part of the country.*


**Hypothesis H3.** 
*Between 2007 and 2013–2014, no progress has been made toward decreasing health inequalities.*


These hypotheses were examined following a preregistered analysis plan (available from https://doi.org/10.17605/OSF.IO/GVYUX accessed on 28 February 2020).

## 2. Material and Methods

### 2.1. Source of Data

Data on the utilization of MHCS from the DHS carried out in the DRC in 2007 and 2013–2014 were used in this study. The DHS is a periodic cross-sectional nationally representative household health survey based on a multi-stage cluster survey design funded by USAID (the U.S. Agency for International Development’s) Bureau for Global Health. Relevant questions related to the utilization of MHCS were retrieved from the women’s questionnaires of both waves, including women of reproductive age (15 to 49 years old) as the study population. Samples selected for enumeration are ensured to be representative and comparative across countries. For the DRC, the DHS involved a two-stage sampling procedure: first selecting the location and then selecting households per location at random. Within a household, respondents were selected by gender for the different questionnaire types. A respondent was included if he/she was a usual member of the household or had spent the night preceding the survey in the household. A random probability sample of households was designed to provide estimates of health, nutrition, water, environmental sanitation, and education at the national level for urban and rural areas and the 11 provinces. The objectives, organization, sample design, and questionnaires used in the DHS surveys are described elsewhere [28].

The global DHS project provided technical assistance in the design, implementation, and analysis of the survey (Monitoring and Evaluation to Assess and Use Results Demographic and Health Surveys: MEASURE DHS) of Macro International, Inc., Irvine, CA, USA.

This study follows the Strengthening the Reporting of Observational Studies in Epidemiology (STROBE) statement: guidelines for reporting observational studies [29]. Detailed descriptions of the application of the STROBE checklist can be found in Appendix A.

### 2.2. Data Collection Procedures

The current study used data from the two most recent rounds (2007 and 2013/2014) of the Congolese Demographic and Health Surveys (EDS-RDC).

The EDS-RDC I (1st wave) [30] was conducted from January to August 2007 using a 2-staged stratified cluster design. It provides data for a wide range of monitoring and impact evaluation indicators on maternal health in the DRC. A total of 9002 households were randomly selected, with a household response rate of 99.3%; 9995 women aged 15–49 were interviewed.

The EDS-RDC II (2nd wave) [31] was implemented from November 2013 to February 2014 using a multi-stage cluster sample survey. A total household sample of 18,360 was randomly selected, with a household response rate of 98.6%; 18,827 women aged 15–49 were interviewed.

Both surveys provide nationally representative maternal and child health estimates and basic demographic and health information [32]. All survey data are presented at both the national and sub-national levels. The latter is often, but not always, provinces or a group of provinces. All of the information collected is representative of the national level, the place of residence (urban and rural), and the level of each of the eleven administrative provinces at the time of the survey. The results also represent the level of each of the twenty-six new administrative provinces. All interviews were administered by the same company, using similar sampling designs and a standard set of questionnaires.

#### Ethics Approval and Consent to Participate

In this study, we made use of secondary DHS data from DRC. Therefore, our study did not require formal ethics approval. All ethics procedures were the responsibility of the institutions that either commissioned, funded, or carried out the original DHS surveys. The Institutional Review Board of Macro International, Inc. reviewed and approved the MEASURE Demographic and Health Surveys Project Phase II in compliance with the United States Department of Health and Human Services requirements for the “Protection of Human Subjects” (45 CFR 46). The 2007 and 2013–2014 EDS-RDC surveys were categorized under that approval. In addition, the study complied with the Maastricht University code of ethics for research in the social and behavioral sciences involving human participants, as well as with the national guidelines.

### 2.3. Variables of the Study

#### 2.3.1. Outcome Variables

In this study, the selection of the primary outcome variables was guided by the framework of indicators proposed by the ‘Countdown to 2030’ global monitoring activities to track universal coverage for reproductive, maternal, newborn, and child health [33] and the WHO Global Reference List of 100 Core Health Indicators [34]. Notably, we examined a diverse set of MHCS utilization variables at different stages of the pregnancy, namely complete antenatal care (ANC), delivery, and postnatal care (PNC). We described the utilization of MHCS to the WHO requirements [35,36], which only consider it a positive pregnancy experience when women (1) receive at least one ANC visit during the first trimester of their pregnancy, (2) have at least eight ANC visits in total throughout their pregnancy, (3) are attended to at delivery by a skilled birth attendant (SBA), and (4) deliver in a health facility.

(1)Antenatal care (ANC), also known as prenatal care services, refers to the total number of women aged 15–49 with a live birth in the five years preceding the survey. In the survey, women were asked whether they had at least three visits for ANC checkups, received at least one TT injection, or underwent the following checkups and tests at least once during antenatal visits—weight, height, blood pressure, blood test, urine test—and whether they received information regarding pregnancy for the last birth during the five years preceding the survey. Undergoing a checkup was classified as timely if done within the first trimester; it was classified as late if done beyond the first trimester; the frequency of ANC visits was defined as adequate or inadequate as per the WHO recommendation—including four or more antenatal visits. This information was used to define full ANC in this study.(2)Care during child delivery (safe delivery) is defined as the deliveries conducted either in a medical institution or at home assisted by a skilled person. The indicator provides information about births attended by skilled health personnel (percentage of births with skilled attendants and by place), institutional delivery (measured as the total number of interviewed women who had one or more live births delivered in a (private or public) health facility), and delivery by cesarean section (measured by the total number of live births to women aged 15–49 years delivered by caesarian section (C-section) in a health facility (private or public)). In the survey, women were asked where their children were born, who assisted during the deliveries, and many other delivery characteristics. This information was collected for the last five years preceding the survey.(3)Postnatal care (PNC)—refers to the total number of women aged 15–49 with a last live birth in the last five years before the survey (regardless of the place of delivery). In the survey, women who had their last birth were asked “if they did have any checkups within 48 h after delivery?” and whether or not the “women underwent any health checkup by a health professional after delivery?” In this study, women who went for a checkup at any health facility within two weeks of delivery are considered to have used postnatal care services.

The majority of the selected outcome variables in this study are binary (yes or no) (e.g., Cesarean section, place of delivery, and professional health assistance during delivery), where 1 indicates the use of the service. Only two outcome variables, number of antenatal visits (0 = no antenatal visits, 1 = 1–3 visits, 2 = 4–7 visits, and 3 = 8 and more visits) and prenatal care received from (0 = no one, 1 = professional care, and 2 = traditional/non-professional care) were categorical variables consisting of three categories.

#### 2.3.2. Independent Variables

At the national level, two independent variables were constructed: the survey year indicates whether a household participated in the DHS survey in 2007 or 2013/2014 as defined by DHS (0 = 2013/2014; 1 = 2007). We described the eastern DRC region (coded as 0) as centered on the North and South Kivu Provinces and nearby Orientale, Maniema, and Katanga. In this region, populations have been living with conflict and displacement for the past two decades due to many years of political and social crisis, and systems providing services for all aspects of life have been weakened. The western DRC region (coded as 1) includes the capital city (Kinshasa) and the provinces of Bandundu, Bas-Congo, Equateur, Kananga, Kasaï Oriental, and Kasaï Occidental, where the burden of the conflict has been less.

All variables that we used in this study are categorized as maternal health services and described in Appendix A.

#### 2.3.3. Control Variables

The included control variables were selected to quantify each determinant’s real contribution to inequality in that specific MHCS utilization variable, such as the type of place of residence of the respondent (living in urban or rural areas was included as dummy variable (2 = rural; 1 = urban); highest education level (a categorical variable was created based on the DRC school system, aggregating education levels as: no education (coded as 0), primary education (coded as 1), secondary education (coded as 2), and higher education (coded as 3); religion (a categorical variable was created using the codes and labels: 1 = catholic; 2 = protestant; 3 = salvation army; 4 = kimbanguist; 5 = other Christian; 6 = muslim; 7 = animist; 8 = no religion and 96 = other); ethnicity (a categorical variable was created using the codes and labels: 1 = bakongo north and south; 2 = bas-kasai and kwilu-kwango; 3 = cuvette centrale; 4 = ubangi and itimbiri; 5 = Uele lake albert; 6 = basele-kivu, Maniema and kivu; 7 = kasai, Katanga, Tanganyika; 8 = Lunda; 9 = pygmy; 96 = others); wealth index (identified five equal categories: poorest—coded as 1; poorer—coded as 2; middle—coded as 3; richer—coded as 4; and richest—coded as 5), and respondents’ current work status was included as a dummy variable (0 = no; 1 = yes) in the datasets.

### 2.4. Statistical Data Analysis

#### 2.4.1. The Magnitude of Inequality (RQ1)

To document the true magnitude of inequality in health, data are required on (i) a measure of health (e.g., health status, health care, and other determinants, and the social and economic consequences of ill health) and (ii) a measure of social position or an ‘equity stratifier’ that defines strata in a social hierarchy (e.g., socioeconomic status, gender, ethnicity, and geographical area) [37]. We assessed the changes in the true magnitude of inequality in utilization of MHCS across different survey years (2007 and 2013–2014) and geographic regions (eastern vs. western of the DRC) using logistic regressions (dichotomous odds ratio (OR) and multinomial (relative risk ratio (RRR)), including previously discussed control variables. Dichotomous logistic regression was chosen for the binary dependent variables, and multinomial logistic regression for the categorical outcome variables. We used logistic regressions to check the adjusted effects of selected socioeconomic and demographic characteristics on the utilization of maternal healthcare services. Logistic regressions allow for the prediction of the relationship between the dependent and independent variables, taking into account multiple control variables. All the independent variables were verified for association with dependent variables at the bivariate level using chi-square tests. We considered *p* ≤ 0.05 as the criterion for statistical significance. Logistic regression analysis results have been presented with 95% confidence intervals (95% CI).

#### 2.4.2. Inequality Distribution (RQ2)

To analyze the distribution of inequality in each selected MHCS utilization variable and every region, we used the Gini coefficient (Gini) and the Lorenz curve. The Lorenz curve is a graphical representation of a function of the cumulative proportion of resources or services of ordered institutions mapped onto the corresponding cumulative proportion of their size. In a Lorenz curve diagram, an unequal distribution of inequality in the utilization of MHCS will loop further down and away from the 45-degree line. In contrast, a more equal distribution in the utilization of MHCS will move the line closer to the 45-degree line.

The Gini is defined as twice the area between the Lorenz curve and the diagonal. It reflects the ratio of the area between the Lorenz curve and the diagonal line to the whole area below the 45-degree line [38]. It ranges from zero (when there is no inequality = perfectly equal distribution) to one (most unequal = when all the population’s health is concentrated in the hands of one person). We used the Gini as a critical measure of inequality for each selected MHCS utilization variable and every region separately (overall and for 2007 and 2013–2014 separately).

#### 2.4.3. Inequality Trends (RQ3)

To investigate inequality trends in the MHCS utilization variables, we used the Wagstaff two group (2007 and 2013–2014 DHS and eastern and western regions) concentration indices (CI) comparison method (using STATA command conindex), which provides point estimates and standard errors of a range of concentration indices [39,40].

We used survey data to classify households into wealth quintiles based on ownership of household assets and housing characteristics. Wealth quintiles represent the relative socioeconomic position of a given country at a specific time rather than absolute wealth, all of which should be account considered when comparing wealth-related inequalities within countries. Thus, wealth quintiles are always a relative measure of how wealth is distributed within the population from the way the quintiles were calculated. For example, wealth quintiles calculated from a survey representative of one specific region of a country will only represent the distribution of wealth in that geographic region.

The DHS Wealth Index is based on the assumption that the possession of assets, services, and amenities is related to the relative economic position of the household in the country [41]. Based on the presence or absence of a large number of potential household assets, the DHS computes a continuous wealth index for each survey. The cut-off points in the wealth index at which to form the quintiles are calculated by obtaining a weighted frequency distribution of households, with the weight being the product of the number of de jure members of the household and the sampling weight of the household [37].

We finally calculated overall and group-specific CI. The CI is the most appropriate measure of health inequality because it meets the three basic requirements of a health inequality index, namely, (i) that it reflects the socioeconomic dimension of inequalities in health, (ii) that it reflects the experiences of the entire population, and (iii) that it is sensitive to changes in the distribution of the population across socioeconomic groups [42]. While the original application of CI was to study income inequality [43], economists have since extended the application of CI to study social inequality in health [42,44,45]. The CI quantifies the extent to which a health service coverage indicator is concentrated among the poorest or the richest. Subsequently, we adopted inference methods developed by Kakwani et al. [44] to test whether these indices are different from zero. We applied the inference test developed by Bishop et al. [46] to test for changes in the CI over time. To estimate the inequality in the utilization of MHCS to the economic condition of women, we fitted the concentration curves (CC). The CC plots shares of ANC, care during child delivery, and PNC against quintiles of the wealth index.

Data management and data analysis were performed by using STATA Version 12.0 (STATA Corp., College Station, TX, USA) and Distributive Analysis/Analyse Distributive (DAD) 4.4 [47].

## 3. Results

### 3.1. Characteristics of the Study Participants

Table 1 presents the descriptive characteristics of all respondents in the DRC by each survey year. It shows that in both survey waves, most respondents came from rural areas (60%), and the proportion of respondents from the western DRC decreased from 67% in 2007 to 62% in 2013–2014. The country’s proportion of respondents with no education decreased from 21% in 2007 to 18% in 2013–2014, while the proportion of respondents with a higher education level slightly increased from 2.91% in 2007 to 2.98% in 2013–2014. Most of the respondents were working, and there was a further increase in the working population from 61% in 2007 to 68% in 2013–2014. The table shows that marital status changed significantly for the respondents living together; it increased from 9.8% in 2007 to 16.9% in 2013–2014, respectively. The country’s dominant ethnic groups are located in the Kasai, Katanga, and Tanganyika (27%), followed by the Basele-Kivu, Maniema, Kivu (20%), and the Bas-Kasai and Kwilu-kwango (15%). Christianity is the most practiced religion (about 29% Catholic, 29% Protestant, and 35% other Christians).

### 3.2. The Magnitude of Inequality (RQ1)

The true magnitude of inequality in the socioeconomic and demographic characteristics of the study respondents on the utilization of MHCS is shown in Table 2 and Table 3. The results indicate that there have been substantial gains in ANC, delivery, and PNC service utilization.

#### 3.2.1. Antenatal Care (ANC)

The majority of women in the DRC received some kind of ANC services (Table 2). Overall, women living in eastern DRC were less likely to have their first ANC visit within the first trimester, less likely to have checkups (at least once) during ANC visits, and less likely to attend four or more ANC visits than those living in western Congo and meet adequate WHO requirements for ANC utilization. On the contrary, women living in western DRC were less likely to receive a check for height and to provide a blood sample. Women from rural areas were more likely to attend four or more ANC visits than those from urban areas. Women belonging to another religious category (other than Christians) were less likely to complete four or more ANC visits, and women with no religious affiliation were less likely to complete more than eight antenatal visits.

Compared to women from Bakongo North and South ethnic groups, women from Bas-Kasai and Kwilu-Kwango were eight-fold more likely to receive a check for weight and height but less likely to provide a blood sample and to receive TT immunization 2. Women from Cuvette Centrale were 14 times more likely to receive prenatal checks, six times for weight, but less likely to provide a blood sample. Women in Ubangi and Itimbiri were 11 times more likely to provide a urine sample; women in Uele Lake Albert were 35 times more likely to receive prenatal checks and 22 times for weight but less likely to provide a blood sample and to receive prenatal information about complications. Women from Basele-Kivu, Maniema, and Kivu were 16 times more likely to receive prenatal care, nine times for weight, and eight times for a urine sample, but less likely to provide a blood sample. Women from Kasai, Katanga, and Tanganyika were 24 times more likely to receive prenatal checks, 7 times more likely to receive a check for weight and urine samples, but less likely to provide blood samples and receive information regarding pregnancy complications. Women in the category other were 40 times more likely to receive prenatal care and prenatal check urine samples.

Compared to women in the poorest group, women in the poorer and middle groups were twice more likely to receive TT immunization 2 but less likely to receive a check of height. Women in the richer and the richest group were threefold to fourfold more likely to provide blood samples but less likely to receive a check for weight and blood pressure. Women with a secondary or higher level of education were twice as likely to receive TT immunization 2, nearly 24 times more likely to receive information regarding complications, and 2 to 60 times more likely to provide blood samples, but less likely to receive a check for weighing and blood pressure, compared to women with no education.

#### 3.2.2. Delivery

Women were found to be assisted during child delivery. In general, they were more likely to receive professional and traditional delivery care when living in western DRC than in eastern DRC. Compared to 2013/2014, women were nearly 0.8 times less likely to deliver by C-section in 2007. Compared to urban areas, women in rural areas were nearly 0.7 times less likely to have their last birth by C-section. Compared to women in the poorest wealth index group, women in the richer and the richest wealth index groups were nearly two times more likely to deliver by C-section. Women with a higher education level were nearly two times more likely to have their last birth by C-section compared to women with no educational background.

Compared to Christians, Kimbanguist women, as women with other religious affiliations, were less likely to deliver by C-section. Women from Bas-Kasai and Kwilu-Kwango, Cuvette Centrale, Ubangi and Itimbiri, Kasai, Katanga, and Tanganyika ethnic groups were less likely to deliver by C-section compared to women from Bakongo ethnic group. Women from Basele-Kivu, Maniema, and Kivu were nearly 2 to 3 times more likely to deliver by C-section.

#### 3.2.3. Postnatal Care (PNC)

Inequality was present during the utilization of PNC. In general, women who visited a health facility received a checkup from a health professional after delivery. Women in rural areas were less likely to receive PNC and less likely to visit a health facility in the last 12 months compared to women in urban areas. Compared to 2013–2014, women were nearly 13 times more likely to get a postnatal checkup but less likely to visit a health facility in the last 12 months than in 2007. Compared to Christians, Kimbanguists and other Christian women were less likely to receive PNC.

Compared to women from the Bakongo North and South ethnic groups, women from Bas-Kasai and Kwilu-Kwango were more likely to receive postnatal checkups. Women were more likely to visit a health facility in the last 12 months in Cuvette Centrale, Ubangi and Itimbiri, Basele-Kivu, Maniema, Kasai, Katanga, Tanganyika, and Lunda but less likely to receive a postnatal checkup. Women in the richer and the richest groups were most likely to receive PNC compared to women in the poorest group. Women who are working were most likely to visit a health facility in the last 12 months, compared to women who are currently not working. Women with a primary, a secondary, or a higher level of education were most likely to receive a postnatal checkup, visit a health facility in the last 12 months, and most likely to receive information regarding the complication, respectively.

Table 3 presents the results of the multinomial regression analysis for the categorical outcome variables.

#### 3.2.4. Number of Antenatal Visits

Women were more than twice as likely to complete four to seven antenatal visits when living in the western DRC compared to the eastern DRC. Women from rural areas were four times more likely to attend eight or more antenatal visits than those from urban areas. Women belonging to another religious category (other than Christians) and women with no religious affiliation were less likely to complete more than eight antenatal visits.

Compared to women from Bakongo North and South ethnic groups, women from Basele-Kivu, Maniema, and Kivu were more than four times more likely to complete 4–7 antenatal visits, and women from Lunda were 33 times more likely to complete more than eight antenatal visits. However, women from Cuvette Centrale were less likely to complete 1–3 antenatal visits.

Compared to women with the poorest wealth index, women with middle, richer, and richest wealth indexes were two and seven times more likely to complete more than eight antenatal visits. Women with primary, secondary, or higher education were 2 to 6 times or 33 times more likely to complete more than eight antenatal visits compared to women with no education.

#### 3.2.5. Prenatal Care Received

Women were twice as likely to receive professional prenatal care and nearly five times more likely to receive traditional prenatal care when living in western DRC than in eastern DRC. Women practicing Islam and women with no religious affiliation were less likely to receive professional prenatal care.

Compared to women from Bakongo North and South ethnic groups, women from Basele-Kivu, Maniema, and Kivu were three times and 14 times more likely to receive professional and traditional prenatal care, respectively. Women from Uele Lake Albert were nearly 17 times more likely to receive traditional prenatal care.

Compared to women with the poorest wealth index, women with the middle wealth index were twice as likely to receive professional and traditional prenatal care. Women with primary or secondary education were more likely to receive professional and traditional prenatal care than women without education.

### 3.3. Inequality Distribution (RQ2)

Table 4 presents the Gini coefficients for each selected MHCS utilization variable between the regions (western vs. eastern DRC) and between the years of the surveys (2007 vs. 2013–2014). It shows that the Gini varied between 0.10 and 0.98, indicating the presence of inequality in both regions and over time, but also considerable heterogeneity between those. Overall, enormous inequality could be observed in prenatal care for (urine samples (0.98), followed by prenatal check number (whether or not having ANC during pregnancy) (0.94); and in delivery by C-sections (every birth (0.91); and last birth by C-section (0.93)). On the contrary, more equality could be observed in the received prenatal care (i.e., number of antenatal visits, TT immunization, received pregnancy information) and in the received postnatal care (i.e., received postnatal checkups and assistance during delivery).

Between 2007 and 2013–2014, data show an overall increase in the Gini for C-sections—particularly in western DRC, where a slight increase was observed in the last birth by C-section (from 0.96 to 0.97)—and for prenatal care from 0.94 to 0.95 in whether or not having ANC during pregnancy, from 0.89 to 0.93 in weight. Similarly, an increase from 0.93 to 0.94 in whether or not having ANC during pregnancy) and from 0.97 to 0.99 for urine samples was observed in eastern DRC. Overall, there was a decrease in the Gini for received prenatal care for tetanus injections and received pregnancy information in eastern DRC but an increase in the Gini for a received postnatal checkup in both geographic regions.

Appendix A displays the Lorenz curves for each MHCS utilization variable separately. The Lorenz curves are relatively far from the line of equality, suggesting a high degree of inequality in the selected MHCS variables. The most significant degree of inequality was observed for prenatal checks for urine samples, whether or not having ANC during pregnancy, height, weight, blood pressure, and C-sections, while the smallest degree of inequality was observed for received prenatal care (i.e., blood samples, TT immunization, the number of antenatal visits, and received pregnancy information), received postnatal checkups, assistance during delivery, and visited the health facilities in the last 12 months.

### 3.4. Inequality Trends (RQ3)

The current analysis found inequality in the utilization of ANC, delivery, and PNC services in DRC—a summary of all results from this analysis is in Table 5.

#### 3.4.1. Antenatal Care (ANC)

Significant differences were found in ANC service utilization between the regions (CI 0.03 in western DRC vs. 0.10 in eastern DRC) and between the years of the surveys. However, patterns of inequality remained relatively consistent for prenatal check numbers, weight, height, and prenatal check blood pressure in both regions. More specifically, in western DRC, a slight decrease could be observed in the CI for the prenatal care received, for prenatal check for a blood sample, and received pregnancy information. No changes could be observed in the CI for tetanus injections and the number of ANC visits. However, in eastern DRC, a slight decrease could be observed in the CI for the prenatal check for height, received pregnancy information, and the number of ANC visits, but a slight increase in tetanus injections. No changes could be observed in the CI for prenatal care received urine sample check and prenatal check blood sample.

#### 3.4.2. Delivery

Between 2007 and 2013–2014, we found a decrease in the CI for delivery by C-section. Particularly in eastern DRC, a decrease could be observed in the CI for both every birth and the last birth by C-section at the same time, while the CI for delivery by C-section remained relatively consistent in western DRC.

#### 3.4.3. Postnatal Care (PNC)

Overall, it could be observed that there was a decrease in the CI for received postnatal checkups and visited health facilities in the last 12 months but a slight increase in the CI for assistance during delivery between 2007 and 2013–2014. For instance, in western DRC, we found a decrease in the CI for the three postnatal variables (received postnatal checkup, visited health facilities in the last 12 months, and assistance during delivery). In eastern DRC, the same trend could be observed in only two variables (received postnatal checkup and visited health facilities in the last 12 months); a slight increase could be observed in assistance during delivery.

## 4. Discussion

This study assessed inequality trends during the utilization of MHCS in post-conflict DRC. While continuous improvements in the utilization of MHCS were found at different stages of pregnancy, several aspects remain inequitable. Moreover, our study found important variations in the utilization of MHCS by geographic region, socioeconomic households, and survey years, shedding light on disparities that need to be addressed. These variations were investigated, and the key results are discussed next.

On the magnitude of inequality, both the odds and the relative risk ratios revealed some degree of inequality during the utilization of MHCS. In the DRC, inequalities could be observed between the western and eastern regions, the poorest and richest socioeconomic groups, and between 2007 and 2013–2014. When zooming in on the levels of utilization of MHCS, the study indicates that these were higher in western compared to eastern DRC; in rural compared to urban areas; among Christians compared to other religious affiliations; in women with a primary, secondary, or higher level of education compared to women with no education; in women from the richer and the richest wealth index; and 2013–2014 as compared to 2007. Our finding that the magnitude of inequality in MHCS utilization is substantial in the DRC is not coincidental. Strong regional inequalities in health have been previously observed within and among countries [48,49,50,51]. In Afghanistan, for instance, one study comparing various provinces based on the severity of conflict showed that the mean coverage of ANC, facility delivery, and SBA was significantly lower for severe conflict provinces when compared to minimal conflict provinces—suggesting that there are notable disparities between provinces [52].

These findings are essential in the context of DRC because the magnitude of inequality in MHCS utilization may be related to the decade-long armed conflict in the country. They put forward the need for designing appropriate programs that aim to increase MHCS utilization, particularly for women belonging to lower economic strata, those belonging to other religious affiliations than Christians, and those living in eastern and rural areas who were less likely to meet the WHO’s requirements of a positive pregnancy experience.

The logistic and multivariate regressions show that ethnicity continues to influence the utilization of MHCS, mainly in the country’s dominant ethnic groups. These findings suggest that there is a consistent pattern of disparities among the different ethnic groups that have been lagging, suggesting that ethnicity could have an essential role in program effectiveness. These findings are consistent with previous studies showing that ethnicity influenced the utilization of maternal health services [53,54,55]. Specifically for the DRC, ethnicity plays a vital role in the acquisition, maintenance, and distribution of wealth [56]—which may influence the utilization of the MHCS.

Over time, inequality in the distribution of MHCS was present in both regions. Total inequality was present in ANC and delivery by C-sections, while some degree of equality could be observed in the received PNC. A few studies have looked at the regional distribution of health indicators within a single country and found that substantial differences among subareas were apparent [57,58], suggesting that inequitable distributions of healthcare services across geographic locations may result in poor or underutilization of MHCS. However, further breakdowns in the distribution of MHCS utilization are needed to explain the differences between subareas. For the DRC, this finding is fundamental because it gives directions for identifying subareas of relatively high need for MHCS utilization.

Regarding inequality trends, a decrease in the utilization of prenatal and postnatal checks and professional assistance during delivery could be observed in respondents from rural areas. This finding suggests that rural locations also accounted for the observed decrease in the utilization of MHCS and is consistent with previous studies showing that utilization of MHCS is lowest in rural areas [59], and the risk of maternal mortality is highest amongst women in rural areas [60].

In this study, we found that the highest educational attainment level was positively associated with utilizing ANC, delivery, and PNC. These findings are consistent with results from previous studies in post-conflict settings showing that maternal education level is a critical aspect in the utilization of MHCS [50,61,62,63]. For the DRC, the few available studies cannot explain whether the association between maternal education and maternal healthcare utilization could be attributed to other factors. Given that SDGs are interdependent, ensure healthy lives, and promote well-being for all, it is only possible if other SDGs, such as SDG 1 (ending poverty), SDG 4 (improving access to education), and SDG 5 (guaranteeing gender equity), among others [64,65], are achieved. The DRC could meet SDG 3.1 (reduce the global maternal mortality ratio to less than 70 per 100,000 live births by 2030) by funding maternal health services and education and developing and maintaining a supportive monitoring process—as both are needed.

Wealth was identified as a significant factor influencing the utilization of MHCS in the DRC. For instance, women with a high wealth index had a higher chance of completing adequate ANC visits and receiving delivery care. Moreover, being currently employed or unemployed also revealed a relation to MHCS utilization. Also, being employed increased the possibility of visiting health facilities in the last 12 months, while being unemployed decreased professional as well as non-professional assistance during delivery. Our findings are consistent with findings from another study conducted in Ghana, Senegal, and Sierra Leone, showing that women with lower wealth did not benefit from the positive effects of the policy reform (e.g., removing user fees) to access facility-based delivery services [66].

We find that each variable of MHCS utilization presents a different pattern, and some variables of MHCS utilization may be more sensitive than others. For example, a decrease could be observed in received postnatal checkups, visited health facilities in the last 12 months, and assistance during delivery in both regions in the DRC. Poorer women or women residing in the eastern DRC have higher levels of inequality in the utilization of MHCS as compared to the richest women or women residing in the western DRC. These findings show persistent patterns of inequality among regional women’s groups and are consistent with previous studies showing a strong positive relationship between wealth and health [42,67,68,69], suggesting that the higher the wealth status of women, the higher their likelihood of seeking appropriate MHCS.

Within-country variations are products of complex socioeconomic factors, showing that no single measure of equality can capture all disparities. In this regard, there might be other factors to consider in future research, such as post-war country status, political orientation, history of dictatorship, and human rights that are not included in the DHS dataset but would highlight more about the influence of maternal healthcare services distribution and utilization in the DRC.

From a policy perspective, our findings provide valuable guidance for policymakers and stakeholders working towards improving MHCS utilization in the DRC and similar contexts.

### Strengths and Limitations

Population-representative data on health status and its determinants are a critical need in the post-conflict context [70]. The use of recent and high-quality data is especially preferable when analyzing maternal healthcare service utilization for data-driven decision-making. However, in several African countries (including the DRC), no national health survey data have been available for several decades. The DHS has several important advantages that make it particularly useful as a programming tool in post-conflict environments.

A major strength of this study is that the DHS produces high-quality data, which is representative of the sub-national regions [32] and provides much-needed data on health service utilization [71]. Given the general lack of primary data in post-conflict settings, we strongly believe that these DHS data are still the most reliable data source that could be used to analyze maternal health service utilization. For these reasons, we used the two most recent DHS data collected in 2007 and 2013–2014 as the primary source of data to assess inequality trends in the utilization of MHCS in the DRC. Since the study uses a high-quality DHS database, the findings are reliable for decision making. Moreover, the DHS creates a unique opportunity to investigate the levels and trends in socioeconomic inequalities in maternal health variables at a scale that was never possible in the past. We were able to disaggregate the DHS data of key health services indicators to assess geographic and socioeconomic characteristics of MHCS utilization. However, given that this analysis is based on secondary data, several limitations to this study must be considered. Firstly, both surveys only included women aged 15–49 years. Knowing the correct age is critical for identifying at-risk women and ascertaining age-specific and age-adjusted risks of maternal mortality. Respondents in DHS surveys are sampled in such a way that the survey sample represents the population (women aged 15–49) of the country. However, the data on siblings are collected from respondents aged 15–49 only, which, by the study design, truncates the siblings in extreme age groups in the reproductive period. For example, a respondent of age 15 is unlikely to have a sibling above age 45, and similarly, a respondent of 49 is unlikely to have a sibling below 20. As a result, the siblings in the age range below 20 or above 45 may not be captured adequately from the DHS survey respondents. Therefore, it is difficult to assess age truncation or underreporting of adult sisters in these two extreme age groups from sibling survival history data [72]. As such, the surveys excluded women below the age of 15 years as well as women above 49 years (and thus, study findings cannot be generalized outside of the sampled population). Because there might be an issue in terms of using MHCS under 15 years, we believe this age range represents women of reproductive age, which is required for our study. Secondly, all the health measures in DHS are collected based on a self-report or proxy report except for height and weight and a few other outcomes, such as anemia. Misclassification biases can occur. Often, its magnitude is also unknown, making correction difficult. For these reasons, we tried to interpret individual-level data more carefully, especially when making causal interpretations. Thirdly, an appropriate variable for work status could have been measured by checking “work status during pregnancy” and “marital status during pregnancy”; however, those variables are not available. Finally, all information collected in DHS surveys (except for weight, height measurements, and vaccination data) is subject to reporting and recall biases that can arise from the recall period or sampling approach. However, a detailed evaluation of DHS data has shown that these data are reasonably well-reported [71], and appropriate strategies are embedded into the design of DHS data collection tools that address recall bias.

## 5. Conclusions and Recommendations

The purpose of this study was to assess inequality trends during the utilization of MHCS in post-conflict DRC. Although it could be argued that there has been a declining trend for some variables from 2007 to 2013–2014, several factors, such as place of residence, ethnicity, education level, religious affiliation, wealth index, and year of survey, were associated with inequality in the utilization of ANC, delivery care, and PNC. Thus, to reduce inequalities in the utilization of MHCS in the DRC, innovative strategies targeting these factors are needed at the regional, subnational, and national levels. Building on our research questions, three key messages emerged from the current analysis: First, substantial gains have been observed in the utilization of ANC, delivery, and PNC services. Second, for some variables, the CCs are far from the line of equality, and the CI are different from zero, suggesting a pro-urban, pro-wealthier, and pro-western DRC distribution. To meet WHO requirements, all women in the DRC should receive at least one ANC visit during the first trimester of their pregnancy, increase to eight ANC visits in total throughout their pregnancy, be attended to at delivery by an SBA, and deliver in a health facility. Third, the current analysis found inequality in the utilization of ANC, delivery, and PNC services in DRC. Trend analyses indicate that region, ethnic group, the place of residence of women, the wealth index, and the level of education of women influence MHCS utilization to some extent. For the DRC, this evidence should serve as a foundation for designing targeted interventions aiming to reduce inequality in MHCS utilization. At the same time, understanding the multiplicity of factors that influence the utilization of MHCS is key to the development of interventions that will work in reducing maternal mortality.

Further research is needed to shed light on the eastern–western and rural–urban differences in not only MHCS utilization but also in the differential factors with significant influence on ANC, delivery, and PNC. Qualitative research on barriers to the utilization of MHC services among poorer, rural, and underserved women is needed to gain insight into inequality trends during the utilization of MHCS in post-conflict settings.

## Figures and Tables

**Table 1 healthcare-11-02871-t001:** Descriptive characteristics of all respondents in the DRC by survey year (2007 and 2013–2014).

Variables	Overall	2007 DHS (N = 14.752)	2013–2014 DHS (N = 27.483)
	(%)	(%)	(%)
Place of residence			
Urban	40.3	47.91	36.26
Rural	59.7	52.09	63.74
Highest Education Level			
No education	18.96	21.08	17.83
Primary	38.51	37.8	38.88
Secondary	39.58	38.21	40.31
Higher	2.96	2.91	2.98
Region			
Kinshasa	12.04	16.67	9.58
Bandundu	11.85	9.42	13.14
Bas-congo	5.81	7.3	5.02
Equateur	12.5	9.07	14.32
Kasai Occidental	7.59	7.27	7.76
Kasai Oriental	10.2	8.66	11.01
Katanga	10.83	9.25	11.66
Maniema	5.93	8.54	4.54
Nord-Kivu	6.84	8.16	6.13
Orientale	10.03	7.55	11.35
Sud-Kivu	6.38	8.07	5.49
Wealth Index			
Poorest	21.75	19.03	23.19
Poorer	19.09	17.64	19.87
Middle	19.05	18.38	19.41
Richer	18.76	20.17	18.01
Richest	21.35	24.78	19.53
Religion			
Catholic	29.14	29.67	28.86
Protestant	28.84	30.7	27.85
Salvation Army	0.24	0.36	0.18
Kimbanguist	3.07	3.23	2.99
Other Christian	34.74	32.33	36.02
Muslim	1.69	1.96	1.54
Animist	0.42	0.52	0.37
No religion	0.92	1.02	0.87
Bundu dia Kongo	0.08		0.12
Vuvamu	0.02		0.03
Other	0.65	0.13	0.93
.	0.19	0.08	0.25
Ethnicity			
Bakongo North and South	10.29	13.69	8.49
Bas-Kasai and Kwilu-Kwango	14.86	12.94	15.88
Cuvette Centrale	9.1	8.23	9.56
Ubangi and Itimbiri	9.81	6.29	11.68
Uele Lake Albert	7.34	4.85	8.66
Basele-k, man. and Kivu	19.75	25.12	16.9
Kasai, Katanga, Tanganika	26.88	26.76	26.95
Lunda	1.01	0.98	1.03
Pygmy	0.22	0.09	0.29
Foreign/non-Congolese	0.32		0.49
Others	0.25	0.71	0.01
.	0.16	0.33	0.07
Currently Working			
No	33.76	38	31.51
Yes	65.99	61.93	68.15
.	0.25	0.07	0.35
Marital Status			
Never married	24.36	24.76	24.14
Married	51.54	56.03	49.15
Living together	14.5	9.86	16.96
Widowed	2.26	2.11	2.34
Divorced	1.94	1.72	2.06
Not living together	5.4	5.51	5.34

Data are presented as N and percentage. “.” represents a missing/unknown subcategory.

**Table 2 healthcare-11-02871-t002:** Odds ratio of all selected maternal health variables by socio-demographic characteristics—logistic regression.

	Delivery		Antenatal Care								Postnatal Care	
Variables	EverbirthCsection	LastbirthCsection	Number	Weighed	Height	Blood Pressure	Urine Sample	Blood Sample	Tetainjectbp2	Receivedinforegcompl	Receivedpostnatcheckup	Visitedhealthfaclast12months
	odds ratio	odds ratio	odds ratio	odds ratio	odds ratio	odds ratio	odds ratio	odds ratio	odds ratio	odds ratio	odds ratio	odds ratio
Base = Eastern Congo												
Western Congo	1025	1236	1444	1277	0.298 ***	1476	3.745 ***	0.623 **	0.841	0.751	0.997	1009
	(0.707–1.487)	(0.886–1.725)	(0.715–2.916)	(0.647–2.521)	(0.147–0.607)	(0.780–2.794)	(1.593–8.805)	(0.400–0.971)	(0.509–1.387)	(0.501–1.126)	(0.837–1.187)	(0.905–1.126)
Base = 2013–2014												
2007	0.793 **	0.823	0.921	1252	0.794	1061	1006	0.949	0.758	0.373 ***	13.06 ***	0.836 ***
	(0.655–0.959)	(0.676–1.001)	(0.527–1.609)	(0.787–1.992)	(0.393–1.604)	(0.737–1.527)	(0.566–1.785)	(0.713–1.262)	(0.539–1.066)	(0.282–0.493)	(7.255–23.51)	(0.777–0.899)
Base = Catholic												
Protestant	0.920	0.819	1005	0.792	1163	0.873	0.705	1270	1035	1201	1032	1046
	(0.720–1.176)	(0.632–1.060)	(0.517–1.952)	(0.461–1.359)	(0.594–2.280)	(0.561–1.359)	(0.361–1.375)	(0.900–1.793)	(0.695–1.541)	(0.855–1.686)	(0.890–1.197)	(0.953–1.148)
Kimbanguist	0.404 ***	0.357 ***	1745	0.204 **	0.145	0.877	0.527	1634	0.820	0.952	0.484 ***	0.811
	(0.221–0.740)	(0.184–0.693)	(0.511–5.959)	(0.0466–0.895)	(0.0181–1.162)	(0.381–2.018)	(0.0634–4.376)	(0.717–3.725)	(0.353–1.905)	(0.485–1.871)	(0.338–0.693)	(0.653–1.008)
Other Christians	0.781 **	0.743 **	1269	0.896	1663	0.806	0.572	1173	1221	0.829	0.783 ***	1033
	(0.619–0.984)	(0.578–0.955)	(0.604–2.666)	(0.507–1.584)	(0.831–3.327)	(0.519–1.252)	(0.295–1.107)	(0.826–1.666)	(0.802–1.860)	(0.591–1.164)	(0.676–0.906)	(0.945–1.130)
Muslim	0.544	0.543	2.753	1570	0.187	0.254	1163	1264	3033	0.600	0.840	0.907
	(0.287–1.033)	(0.274–1.074)	(0.882–8.601)	(0.550–4.482)	(0.0194–1.802)	(0.0350–1.837)	(0.138–9.828)	(0.467–3.426)	(0.698–13.18)	(0.159–2.259)	(0.527–1.340)	(0.692–1.190)
Animist	0.529	0.574	4291			1015		1419	1603	2043	0.926	1421
	(0.0861–3.254)	(0.0938–3.512)	(0.646–28.50)			(0.190–5.428)		(0.396–5.081)	(0.435–5.904)	(0.533–7.832)	(0.438–1.956)	(0.888–2.272)
No religion	0.517	0.462		0.254		0.567	2172	3.945 **	1554	1215	0.802	1024
	(0.167–1.601)	(0.118–1.810)		(0.0481–1.339)		(0.109–2.949)	(0.479–9.852)	(1.146–13.58)	(0.380–6.350)	(0.418–3.530)	(0.485–1.326)	(0.692–1.513)
Other	1137	1124	1571	3.293	2679	0.889		0.341	0.661	1154	0.551 ***	0.960
	(0.472–2.741)	(0.436–2.900)	(0.291–8.483)	(0.824–13.16)	(0.719–9.983)	(0.183–4.324)		(0.0820–1.416)	(0.185–2.356)	(0.401–3.319)	(0.357–0.852)	(0.697–1.321)
Base = Bakongo North and South												
Bas-Kasai and Kwilu-Kwngo	0.595 ***	0.493 ***	8.754	7.987 ***	7.777 **	0.641	3430	0.475 **	0.425 **	0.650	1.267 **	0.985
	(0.410–0.863)	(0.341–0.713)	(0.879–87.13)	(1.879–33.95)	(1.315–45.98)	(0.307–1.340)	(0.766–15.37)	(0.259–0.870)	(0.187–0.966)	(0.380–1.111)	(1.001–1.604)	(0.866–1.122)
Cuvette Centrale	0.560 **	0.463 ***	13.85 **	5.978 **	1653	1284	2920	0.380 ***	0.593	0.951	0.582 ***	1.321 ***
	(0.335–0.936)	(0.265–0.809)	(1.559–123.1)	(1.146–31.19)	(0.244–11.22)	(0.565–2.920)	(0.560–15.22)	(0.188–0.767)	(0.250–1.407)	(0.477–1.893)	(0.440–0.771)	(1.128–1.548)
Ubangi and Itimbiri	0.535 ***	0.435 ***	6162	1044	2512	0.981	11.64 ***	0.537	0.840	0.631	0.435 ***	1.335 ***
	(0.362–0.791)	(0.285–0.665)	(0.680–55.83)	(0.222–4.909)	(0.417–15.14)	(0.465–2.068)	(2.791–48.53)	(0.285–1.014)	(0.355–1.991)	(0.351–1.135)	(0.338–0.560)	(1.158–1.541)
Uele Lake Albert	1360	1434	34.34 ***	21.52 ***	1027	1389	0.401	0.167 ***	0.487	0.395 **	0.806	1022
	(0.806–2.294)	(0.855–2.404)	(3.445–342.3)	(4.270–108.5)	(0.127–8.335)	(0.481–4.013)	(0.0333–4.820)	(0.0752–0.369)	(0.174–1.367)	(0.187–0.834)	(0.590–1.101)	(0.842–1.240)
Basele-k, man. And Kivu	2.111 ***	2.372 ***	16.19 **	9.505 ***	4289	1286	7.827 **	0.215 ***	0.490	0.675	1143	1.433 ***
	(1.322–3.371)	(1.548–3.636)	(1.623–161.6)	(1.968–45.90)	(0.694–26.51)	(0.479–3.455)	(1.463–41.88)	(0.103–0.449)	(0.193–1.246)	(0.342–1.329)	(0.854–1.531)	(1.215–1.690)
Kasai, Katanga, Tanganika	0.630 ***	0.611 ***	23.73 ***	7.352 ***	3202	1463	6.586 ***	0.219 ***	0.504	0.515 **	0.675 ***	1.235 ***
	(0.447–0.888)	(0.432–0.863)	(2.773–203.0)	(1.781–30.36)	(0.557–18.42)	(0.739–2.895)	(1.577–27.51)	(0.123–0.393)	(0.229–1.107)	(0.308–0.859)	(0.536–0.849)	(1.096–1.393)
Lunda	1486	1618	4774		0.600	0.375		1265	0.768	0.462	1102	1.712 ***
	(0.649–3.400)	(0.693–3.777)	(0.239–95.49)	(0.755–33.42)	(0.0410–8.780)	(0.0744–1.893)		(0.389–4.121)	(0.148–3.983)	(0.120–1.774)	(0.608–1.998)	(1.205–2.432)
Other	0.458	0.523	39.47 ***	1703	5772		41.64 ***	0.214	1728	0.491	0.705	1083
	(0.166–1.266)	(0.189–1.444)	(2.957–526.8)	(0.124–23.40)	(0.375–88.82)		(4.806–360.8)	(0.0407–1.124)	(0.178–16.74)	(0.133–1.816)	(0.371–1.342)	(0.737–1.592)
Base = Urban												
Rural	0.788	0.755 **	1232	1088	1478	1.897 ***	0.717	0.631 **	0.541 **	0.721	0.641 ***	0.884 **
	(0.610–1.018)	(0.579–0.985)	(0.569–2.668)	(0.551–2.147)	(0.706–3.091)	(1.173–3.068)	(0.337–1.524)	(0.440–0.904)	(0.333–0.880)	(0.500–1.038)	(0.547–0.751)	(0.794–0.984)
Base = Poorest												
Poorer	1013	1156	0.678	0.885	0.752	0.910	0.938	1.407	1.547 **	1078	1112	1.130 **
	(0.729–1.407)	(0.813–1.644)	(0.375–1.224)	(0.517–1.514)	(0.351–1.609)	(0.575–1.438)	(0.433–2.032)	(0.959–2.064)	(1.007–2.377)	(0.751–1.549)	(0.947–1.306)	(1.011–1.264)
Middle	1170	1301	0.823	0.988	0.420 **	0.885	1144	1.407	1.691 **	1220	1.236 **	1090
	(0.838–1.634)	(0.913–1.854)	(0.464–1.459)	(0.554–1.762)	(0.194–0.911)	(0.555–1.413)	(0.558–2.346)	(0.972–2.037)	(1.086–2.632)	(0.846–1.759)	(1.049–1.457)	(0.973–1.221)
Richer	1.755 ***	1.726 ***	0.298 **	0.468	0.965	0.599	0.601	2.710 ***	1325	1372	1.401 ***	1.125
	(1.244–2.477)	(1.217–2.447)	(0.112–0.790)	(0.216–1.011)	(0.432–2.156)	(0.350–1.026)	(0.224–1.615)	(1.753–4.189)	(0.794–2.213)	(0.887–2.122)	(1.156–1.697)	(0.988–1.281)
Richest	1.711 ***	1.757 ***	0.105 **	0.143 **	0.436	0.394 **	0.709	4.180 ***	1294	1218	1.824 ***	1126
	(1.173–2.497)	(1.183–2.610)	(0.0179–0.623)	(0.0247–0.825)	(0.107–1.772)	(0.166–0.935)	(0.213–2.361)	(2.254–7.749)	(0.613–2.731)	(0.709–2.091)	(1.444–2.304)	(0.971–1.306)
Base = no—currently not working												
Yes—currently working	0.956	1027	1192	1626	0.612	0.831	0.931	1045	0.789	1007	0.894	1.502 ***
	(0.769–1.189)	(0.826–1.277)	(0.672–2.114)	(0.909–2.908)	(0.341–1.101)	(0.564–1.225)	(0.498–1.741)	(0.761–1.436)	(0.538–1.158)	(0.756–1.341)	(0.786–1.017)	(1.393–1.621)
Base = no education												
Primary education level	0.827	0.799	1070	0.861	1225	0.854	1185	1128	1184	0.827	1.170 **	1.202 ***
	(0.624–1.098)	(0.597–1.069)	(0.615–1.862)	(0.511–1.450)	(0.648–2.316)	(0.562–1.297)	(0.558–2.518)	(0.806–1.578)	(0.802–1.747)	(0.591–1.157)	(1.006–1.359)	(1.084–1.332)
Secondary education level	1083	1043	0.410 **	0.327 ***	0.711	0.833	0.834	2.231 ***	1.702 **	1322	1.574 ***	1.309 ***
	(0.788–1.487)	(0.760–1.431)	(0.195–0.862)	(0.161–0.665)	(0.293–1.726)	(0.500–1.387)	(0.357–1.944)	(1.492–3.336)	(1.041–2.783)	(0.885–1.974)	(1.326–1.869)	(1.167–1.468)
Higher education level	1.669	1.860 **				0.0750 **		59.56 ***	4726	23.93 ***	3.805 ***	1.584 ***
	(0.937–2.973)	(1.033–3.352)				(0.00841–0.669)		(6.627–535.2)	(0.514–43.43)	(3.161–181.2)	(2.247–6.443)	(1.284–1.955)
Constant	0.0823 ***	0.0633 ***	0.00429 ***	0.0150 ***	0.0353 ***	0.181 ***	0.00660 ***	2.838 **	10.85 ***	2.956 **	1216	0.310 ***
	(0.0450–0.150)	(0.0350–0.114)	(0.000318–0.0580)	(0.00210–0.107)	(0.00401–0.311)	(0.0559–0.585)	(0.000916–0.0475)	(1.120–7.189)	(3.345–35.17)	(1.219–7.168)	(0.834–1.773)	(0.247–0.389)
Observations	16.609	16.592	2.163	2.164	2.146	2.184	2.11	2.198	2.169	2.173	11.37	28.643

*** *p* < 0.01, ** *p* < 0.05.

**Table 3 healthcare-11-02871-t003:** Relative risk ratio on categorical variables—multinomial regressions.

Variables	Number Antenatal Visits				Prenatal Care Received from		
	No Visits	1–3 Visits	4–7 Visits	8 or More Visits	Base = No One	Professional Care	Traditional Care
	Relative Risk Ratio	Relative Risk Ratio	Relative Risk Ratio	Relative Risk Ratio	Relative Risk Ratio	Relative Risk Ratio	Relative Risk Ratio
Base = Eastern Congo							
Western Congo		1.577	2.372 ***	1290		1.746 **	5.205 ***
		(0.953–2.609)	(1.406–4.002)	(0.381–4.372)		(1.099–2.775)	(2.124–12.76)
Base = 2013–2014							
2007		0.734	0.711	1010		0.784	0.565
		(0.497–1.082)	(0.475–1.064)	(0.427–2.389)		(0.543–1.133)	(0.257–1.240)
Base = Catholic							
Protestant		0.919	0.926	0.422		0.839	1156
		(0.566–1.494)	(0.565–1.518)	(0.139–1.280)		(0.525–1.341)	(0.432–3.092)
Kimbanguist		0.707	0.437	0.185		0.519	0.427
		(0.304–1.647)	(0.181–1.055)	(0.0246–1.387)		(0.233–1.156)	(0.0809–2.254)
Other Christians		0.876	0.882	0.604		0.870	0.662
		(0.544–1.410)	(0.543–1.432)	(0.209–1.747)		(0.551–1.373)	(0.256–1.710)
Muslim		0.396	0.454	0 ***		0.290 **	4015
		(0.107–1.457)	(0.115–1.794)	(0–0)		(0.0859–0.979)	(0.612–26.34)
Animist		0.347	1441	0 ***		0.714	0.730
		(0.0394–3.060)	(0.111–18.79)	(0–0)		(0.0851–5.982)	(0.0373–14.27)
No religion		0.240 **	0.263 **	0.0250 ***		0.233 ***	0 ***
		(0.0779–0.737)	(0.0841–0.824)	(0.00236–0.265)		(0.0894–0.607)	(0–0)
Other		0.700	0.158 ***	0 ***		0.348 **	1477
		(0.265–1.850)	(0.0422–0.592)	(0–0)		(0.132–0.918)	(0.284–7.694)
Base = Bakongo North and South							
Bas-Kasai and Kwilu-Kwango		0.916	1097	0.813		1034	3614
		(0.335–2.504)	(0.395–3.047)	(0.0694–9.514)		(0.385–2.772)	(0.552–23.68)
Cuvette Centrale		0.313 **	0.819	9.33		0.568	3118
		(0.116–0.847)	(0.301–2.228)	(0.862–100.9)		(0.222–1.455)	(0.519–18.72)
Ubangi and Itimbiri		0.553	0.789	4036		0.668	1078
		(0.204–1.495)	(0.287–2.169)	(0.375–43.48)		(0.256–1.741)	(0.133–8.756)
Uele Lake Albert		0.861	1631	3714		1083	17.07 **
		(0.258–2.873)	(0.462–5.750)	(0.245–56.40)		(0.342–3.431)	(1.890–154.1)
Basele-k, man. And Kivu		2387	4.415 **	8997		3.028 **	13.41 **
		(0.775–7.352)	(1.378–14.15)	(0.650–124.5)		(1.024–8.952)	(1.595–112.7)
Kasai, Katanga, Tanganika		0.483	0.618	4430		0.560	1912
		(0.194–1.202)	(0.243–1.574)	(0.475–41.29)		(0.231–1.354)	(0.319–11.44)
Lunda		1000	2222	33.47 **		1602	0 ***
		(0.210–4.757)	(0.473–10.43)	(1.741–643.4)		(0.382–6.718)	(0–1.07 × 10^−10^)
Other		0.531	0.741	0 ***		0.572	4955
		(0.0966–2.921)	(0.114–4.812)	(0–0)		(0.110–2.984)	(0.247–99.41)
Base = Urban							
Rural		0.719	0.690	3.636 **		0.726	2.576
		(0.404–1.281)	(0.390–1.222)	(1.207–10.95)		(0.425–1.242)	(0.976–6.798)
Base = Poorest							
Poorer		0.941	1300	1851		1169	0.769
		(0.608–1.457)	(0.829–2.037)	(0.646–5.303)		(0.777–1.760)	(0.340–1.741)
Middle		1.699 **	1.928 **	2317		1.959 ***	0.387
		(1.046–2.761)	(1.158–3.210)	(0.747–7.193)		(1.231–3.116)	(0.144–1.040)
Richer		1342	1.928 **	0.825		1.585	1365
		(0.753–2.395)	(1.075–3.458)	(0.142–4.795)		(0.932–2.695)	(0.509–3.657)
Richest		0.684	1970	7.077 **		1257	2451
		(0.233–2.010)	(0.683–5.682)	(1.225–40.90)		(0.455–3.478)	(0.492–12.22)
Base = no—currently not working							
Yes—currently working		1029	1129	0.571		1060	1749
		(0.677–1.565)	(0.747–1.705)	(0.221–1.477)		(0.719–1.562)	(0.802–3.814)
Base = no education							
Primary education level		1.481	1.752 ***	1594		1.512 **	3.554 ***
		(0.995–2.205)	(1.149–2.671)	(0.604–4.209)		(1.037–2.204)	(1.567–8.061)
Secondary education level		3.285 ***	5.633 ***	3.828 **		4.497 ***	4.643 ***
		(1.769–6.100)	(3.039–10.44)	(1.152–12.72)		(2.476–8.165)	(1.647–13.09)
Higher education level		2674	3784	32.45 **		3646	0 ***
		(0.234–30.59)	(0.416–34.44)	(1.435–733.6)		(0.411–32.35)	(0–0)
Constant		3.441 *	1081	0.0111 ***		4.264 **	0.00446 ***
		(0.919–12.88)	(0.277–4.222)	(0.000571–0.215)		(1.228–14.81)	(0.000293–0.0677)
Observations	2.537	2.537	2.537	2.537	2.568	2.568	2.568

Relative risk measures the association between the exposure and the outcome. Robust ci in parentheses (figures in brackets show 95 percent confidence intervals). *** *p* < 0.01, ** *p* < 0.05, * *p* < 0.1.

**Table 4 healthcare-11-02871-t004:** Gini coefficients for all selected maternal variables.

	Eastern DRC			Western DRC		
All Selected Variables	Overall	2007	2013/14	Overall	2007	2013/14
Cesarean-section						
Ever birth C-section	0.91	0.94	0.90	0.96	0.96	0.96
Last birth C-section	0.93	0.95	0.92	0.97	0.96	0.97
Prenatal care						
Prenatal check_no	0.94	0.93	0.94	0.95	0.95	0.95
Received prenatal care	0.17	0.24	0.13	0.15	0.14	0.15
Prenatal check weighed	0.89	0.88	0.88	0.93	0.92	0.94
Prenatal check height	0.93	0.97	0.91	0.97	0.97	0.98
Prenatal check blood pressure	0.84	0.83	0.84	0.82	0.82	0.81
Prenatal check urine sample	0.98	0.97	0.99	0.94	0.95	0.94
Prenatal check blood sample	0.45	0.45	0.45	0.42	0.42	0.42
Tetanus injections	0.18	0.24	0.15	0.17	0.19	0.17
Received pregnancy information	0.45	0.61	0.39	0.51	0.67	0.43
Postnatal Care						
Received postnatal checkup	0.48	0.04	0.49	0.51	0.13	0.52
Visited health facilities in the last 12 months	0.62	0.66	0.61	0.64	0.67	0.62
Assistance during delivery	0.15	0.17	0.14	0.16	0.17	0.15

**Table 5 healthcare-11-02871-t005:** Concentration indices by selected maternal variables, by survey year (2007 vs. 2013–2014), and by geographic regions (western vs. eastern) of the DRC.

	2007				2013/14				Western Congo				Eastern Congo			
All Selected Variables	Group 0 = Eastern Congo	Group 1 = Western Congo	Socioeconomic Inequality in the Health Variable	Statistical Significance Between the 2 Groups in the Socioeconomic Inequality	Group 0 = Eastern Congo	Group 1 = Western Congo	Socioeconomic Inequality in the Health Variable	Statistically Significance between the 2 Groups in the Socioeconomic Inequality	CI	PeriodSurvey = 0 (2013/2014)	PeriodSurvey = 1 (2007)	Test for Stat. Significant Differences	CI	PeriodSurvey = 0 (2013/2014)	PeriodSurvey = 1 (2007)	Test for Stat. Significant Differences
Delivery																
Ever C-section	0.67	0.68	0.68	0.97	0.65	0.68	0.67	0.89	0.68	0.68	0.68	0.97	0.65	0.65	0.67	0.97
Last birth C-section	0.68	0.69	0.68	1.00	0.66	0.69	0.68	0.82	0.69	0.69	0.69	0.94	0.66	0.66	0.68	0.87
Prenatal care																
Prenatal care received	0.47	0.85	0.63	0.47	0.53	0.67	0.59	0.75	0.73	0.67	0.85	0.80	0.52	0.53	0.47	0.83
Prenatal check number	0.67	0.68	0.68	0.97	0.68	0.68	0.68	0.92	0.68	0.68	0.68	0.87	0.67	0.68	0.67	0.91
Prenatal check weighed	0.64	0.67	0.66	0.90	0.65	0.67	0.67	0.82	0.67	0.67	0.67	0.88	0.65	0.65	0.64	0.95
Prenatal check height	0.69	0.69	0.69	0.98	0.66	0.69	0.68	0.54	0.69	0.69	0.69	0.93	0.67	0.66	0.69	0.55
Prenatal check blood pressure	0.59	0.59	0.59	0.71	0.61	0.58	0.59	0.90	0.58	0.58	0.59	0.86	0.60	0.61	0.59	0.74
Prenatal check urine sample	0.69	0.68	0.69	0.53	0.70	0.67	0.68	0.46	0.68	0.67	0.68	0.67	0.70	0.70	0.69	0.91
Prenatal check blood sample	0.22	0.29	0.26	0.44	0.21	0.24	0.23	0.58	0.26	0.24	0.29	0.79	0.21	0.21	0.22	0.27
Tetanus injections	0.53	0.58	0.56	0.88	0.62	0.59	0.60	0.74	0.59	0.59	0.58	0.93	0.60	0.62	0.53	0.67
Received pregnancy information	0.29	0.31	0.31	0.02	0.26	0.21	0.23	0.71	0.03	0.21	0.31	0.24	0.10	0.26	0.29	0.01
Number antenatal visits	0.49	0.11	0.31	0.91	0.13	0.12	0.12	0.72	0.11	0.12	0.11	0.63	0.23	0.13	0.49	0.75
Postnatal care																
Received postnatal checkup	0.70	0.65	0.66	0.49	0.09	0.02	0.03	0.28	0.01	0.02	0.65	0.00	0.10	0.09	0.70	0.00
Visited health facilities in the last 12 months	0.37	0.38	0.37	0.93	0.25	0.30	0.28	0.37	0.33	0.30	0.38	0.71	0.29	0.25	0.37	0.83
Assistance during delivery	0.40	0.49	0.46	0.88	0.49	0.46	0.47	0.88	0.47	0.46	0.49	0.97	0.46	0.49	0.40	0.78

## Data Availability

This study used datasets available from USAID’s Demographic and Health Survey (DHS) program. After registration on the website, datasets can be downloaded and used via the DHS program website: https://dhsprogram.com/data/new-user-registration.cfm (accessed on 21 April 2023).

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
