# Peer review of "Maternal Health Care Service Utilization in the Post-Conflict Democratic Republic of Congo: An Analysis of Health Inequalities over Time"

_healthcare, 2023, doi:10.3390/healthcare11212871_

Round 1

Reviewer 1 Report

Comments and Suggestions for Authors

Comment for the Authors

General Comments:

The manuscript takes on an undeniably crucial topic, i.e., Maternal Health Care Services (MHCS) utilization in emergencies, particularly in the Democratic Republic of Congo. Your work dovetails nicely with the growing body of literature that seeks to address health disparities in conflict and emergency zones. However, there are some pivotal issues that need attention to enhance the manuscript's rigor and relevance.

Scope and Aim: The manuscript successfully bridges an existing gap in our understanding of maternal healthcare service utilization in conflict-affected areas, with an emphasis on the DRC. However, the presentation of the objectives could be more specific to help focus the study further.

Theoretical Framework: While the study alludes to Andersen's Behavioural Model of Health Care Utilization, the conceptual framework's operationalization and integration into the study appear somewhat tenuous. It would be beneficial to elaborate on how this theory guided your research design and interpretation of findings.

Comparative Analysis: The discussion could be enriched by making comparisons with other post-conflict regions. This would help the reader understand whether the observed phenomena are unique to the DRC or could be generalized to other settings.

Policy Recommendations: The manuscript could be more impactful by providing explicit policy recommendations based on the findings. Given the nuanced understanding developed in the article, this is a missed opportunity to articulate actionable next steps.

Clarity and Language: The manuscript is generally well-written, but there are places where the text could be streamlined for clarity. For example, the introduction has a rich background but is a bit convoluted in presenting the key research questions and hypotheses.

Specific Comments:

Introduction:

Referencing SDGs: While the Sustainable Development Goals (SDGs) are mentioned, their contextual relevance could be made more explicit. The manuscript would benefit from a concise explication of how it contributes to the ongoing discourse and policies targeted at achieving SDG 3.

Conflict and Maternal Health: The manuscript does an excellent job highlighting the adverse impact of conflict on maternal health. It may further enrich the narrative by incorporating references to contemporary studies that have explored this relationship.

Clarification of Hypotheses: The hypotheses are clearly stated but may benefit from more rigorous justification rooted in extant literature or preliminary data.

Methods:

Data Source: The use of DHS (Demographic and Health Surveys) data is methodologically sound. However, the limitations related to self-reported data and recall bias should be acknowledged.

Sampling Strategy: More details regarding the sampling strategy would be beneficial. In addition, if stratified sampling was used, elucidating the stratification criteria will offer more comprehensive insights.

Ethical Considerations: The manuscript mentions that all ethics procedures were the responsibility of the original DHS surveys. While this is noteworthy, a separate section discussing the ethical considerations specific to this study, especially considering its sensitive subject matter, would add value.

Variable Selection: It's commendable that the outcome variables were selected based on established frameworks like the 'Countdown to 2030' and WHO guidelines. Still, the manuscript could benefit from a more in-depth discussion of why these specific indicators were chosen.

Results:

Presentation of Results: The results are extensively detailed, but the presentation is overwhelming and hard to digest. A more focused analysis, potentially incorporating data visualization elements like figures and graphs, may enhance clarity and impact.

Inequality Measures: You used the Gini coefficient to measure inequality, which is commendable. However, did you consider alternative measures, such as the Theil index or the Atkinson index, to validate your findings?

Discussion:

On the topic of inequalities, the study could be strengthened by integrating literature on social determinants of health to provide a more comprehensive view.

The connection between the decade-long armed conflict and health disparities needs to be discussed in greater depth, possibly utilizing references to trauma-informed care literature.

Clarification is needed on whether the identified disparities in healthcare services are a result of availability or utilization, as this has significant implications for policy.

Limitations:

The manuscript should further discuss the limitations of using DHS data, particularly the risk of recall bias and the limitations of self-reported data.

It would be helpful to have a more detailed explanation about why the study focused on women aged 15-49, as opposed to including a broader age range.

Conclusion and Recommendations:

The recommendations could benefit from being more specific and targeted. For instance, identifying particular interventions that have been shown to be effective in similar contexts could make the conclusions more actionable.

Comments on the Quality of English Language

Clarity and Language: The manuscript is generally well-written, but there are places where the text could be streamlined for clarity. For example, the introduction has a rich background but is a bit convoluted in presenting the key research questions and hypotheses.

Reviewer 2 Report

Comments and Suggestions for Authors

I congratulate the authors for this thoroughly and well-written study. The topic is highly relevant.

These are my comments:

Use of abbreviations: 

Do not use abbreviations in the abstract. Some scholars who just want to review the abstract will have difficulty understanding it.

Do not use the same abbreviation for different definitions: For example:

CI= concentration indices

CI= confidence interval

Introduction

On page 2, the authors mentioned that they intend to present a longitudinal perspective. However, their data are cross-sectional.

Materials and Methods

Data are quite old. The most recent data are in 2014 (9 years ago) and there are some considerable gaps in years between 2007 and 2013. It would be nice if the authors give some explanations about these two concerns and mention these in the Limitations section. 

Providing a Table for the descriptive statistics of the dependent variables in the body of the paper would be nice.

I do not think work status “currently working” and “marital status” are relevant variables because the survey questioned women about their prenatal, delivery, and postnatal care that occurred 5 years preceding the survey year. These women may not have been working of have a different marital status when they were pregnant. So current work  status and marital status will not affect pregnancy and pregnancy care that occurred 5 years ago. A variable : “work status during pregnancy”, and “marital status during pregnancy” would be more helpful. If those variables are not available, then mentioning that in the Limitations section would be helpful.

Results

In Table 2: the following religion categories are missing: Salvation Army, Bundu dia kongo, and Vuvamu are not reported. Some ethnic categories are also missing (ie. pygmy, foreign-non-congolese).

Comments on the Quality of English Language

The English is very well-written.

Reviewer 3 Report

Comments and Suggestions for Authors

REVIEW REPORT FOR THE STUDY “MATERNAL HEALTH CARE SERVICES UTILIZATION IN THE POST-CONFLICT DEMOCRATIC REPUBLIC OF CONGO: ANALYSIS OF HEALTH INEQUALITIES OVER TIME”

Journal: Healthcare

The paper " Maternal Health Care Services Utilization in the Post-conflict Democratic Republic of Congo: Analysis of health inequalities over time", performs an study on inequality in maternal healthcare services utilization in Democratic Republic of Congo, using Demographic and Health Surveys, 2007 and 2013-2014.

Title and summary. The title and abstract express well the object of study, objectives, and results of the article.

Structure of the article. The contents are well organized and they adhere to the IMRaD structure. It includes a theoretical framework of the research problem but at this point, I suggest the authors incorporate some other bibliographic references that I miss in the text:

Misu, F., Alam, K. Comparison of inequality in utilization of maternal healthcare services between Bangladesh and Pakistan: evidence from the demographic health survey 2017–2018. Reprod Health 20, 43 (2023). https://doi.org/10.1186/s12978-023-01595-y

Novignon, J., Ofori, B., Tabiri, K.G. et al. Socioeconomic inequalities in maternal health care utilization in Ghana. Int J Equity Health 18, 141 (2019). https://doi.org/10.1186/s12939-019-1043-x

Smith, M. J. (2015). Health equity in public health: Clarifying our commitment. Public Health Ethics, 8(2), 173-184.

Focusing on the opportunity of the study, it must be said that it is useful work since it covers one of the major problems resulting from a health care system.

Materials and methods.

Regarding the material and methods section, the methodology is tailored to the object of study and the objectives and is explained in a transparent manner while it has been validly applied to guarantee the results.

However, I would like to suggest to the authors, incorporate definitions so that readers know exactly what authors mean by inequity as there is great diversity in definitions of the concept of health equity and such implications are widely discussed (Whitehead 1992; Braveman and Gruskin 2003; Braveman, Cubbin et al. 2005). Sometimes this notion, often defined through its opposite, which is inequity or inequality, refers to the description of avoidable or remediable differences between different groups of people (Whitehead 1992; Kawachi, Subramanian et al. 2002), and thus includes a moral and ethical dimension (Daniels 2006; Smith 2015). Sometimes, it refers to the consequences of a public health intervention. Indeed, in the latter dimension, the notion of inequity implies an intervention that will increase already existing health inequalities. These two notions sometimes merge, but often the implications and assumptions for underlying social justice theories are rarely discussed (Smith 2015).

Results.

The results are significant and they are presented in an adequate and understandable way not only through narration but also with self-explained tables and figures that are also well elaborated in terms of presentation. The results justify and relate to the objectives and methods and the results are of sufficient interest.

Discussion.

The discussion appropriately compares the study results with other works, highlighting the main study findings. However, I would propose the inclusion of three bibliographic references in the discussion section:

McKinnon, B., Harper, S. et Kaufman, J. S. (2015). Who benefits from removing user fees for facility-based delivery services? Evidence on socioeconomic differences from Ghana, Senegal and Sierra Leone. Social Science et Medicine, 135, 117-123.

Balhasan Ali, Paramita Debnath, Tarique Anwar, Inequalities in utilisation of maternal health services in urban India: Evidences from national family health survey-4, Clinical Epidemiology and Global Health, Volume 10, 2021,

http://doi.org/10.1016/j.cegh.2020.11.005.

Sanni Yaya and Bishwajit Ghose. Global Inequality in Maternal Health Care Service Utilization: Implications for Sustainable Development Goals. Health Equity.Jul 2019.145-154. http://doi.org/10.1089/heq.2018.0082

Bibliography.

The 19% of the bibliography cited in the study belongs to the previous five years.

Overall, it is an interesting study and should be considered for publication in Healthcare, once the minor revisions proposed have been resolved.

Reviewer 4 Report

Comments and Suggestions for Authors

REVIEWED REPORT ON A MANUSCRIPT TITLED: MATERNAL HEALTH CARE SERVICES UTILIZATION IN THE POST-CONFLICT DEMOCRATIC REPUBLIC OF CONGO: ANALYSIS OF HEALTH INEQUALITIES OVER TIME.

Introduction:

The manuscript titled "Maternal Health Care Services Utilization in the Post-conflict Democratic Republic of Congo: Analysis of Health Inequalities" provides an in-depth analysis of the utilization of maternal health care services in the challenging context of post-conflict Democratic Republic of Congo (DRC).

Aim of the Paper:

The primary aim of this paper is to assess inequality trends during the utilization of MHCS in DRC after a period of conflict and turmoil. Specifically, the authors aim to understand the factors influencing maternal health care utilization and to analyze the existing health inequalities in access to these services. This research is of critical importance as it addresses a pressing issue in a region that has faced significant challenges in recent history.

Main Contributions:

The manuscript makes several notable contributions to the field of maternal health care services and health inequalities:

Comprehensive Data Analysis: The authors have conducted a thorough analysis of data collected from the DRC, employing statistical methods to examine the utilization patterns of maternal health care services. This rigorous analysis provides valuable insights into the factors affecting access to care.

Post-Conflict Context: The study focuses on the post-conflict period, which is a unique and under-researched context. This focus allows for a better understanding of the challenges and opportunities in rebuilding healthcare systems and addressing health inequalities in such settings.

Policy Implications: The manuscript discusses policy implications derived from the findings, which is crucial for informing healthcare policy and interventions in post-conflict regions. It provides valuable guidance for policymakers and stakeholders working towards improving maternal health care in the DRC and similar contexts.

Health Inequality Analysis: The paper effectively highlights the existing health inequalities in maternal health care service utilization, shedding light on disparities that need to be addressed. This information can serve as a foundation for designing targeted interventions.

Strengths:

The manuscript possesses several strengths that enhance its contribution to the field:

Robust Methodology: The research design and methodology employed in this study are robust, ensuring the reliability of the findings. The authors have used nationally representative data, strengthening the generalizability of their results.

Clarity of Presentation: The paper is well-structured and well-written, making it accessible to both researchers and policymakers. The logical flow of the manuscript aids in understanding the complex issues under investigation.

Policy Relevance: The discussion of policy implications and recommendations makes the research directly applicable to real-world efforts to improve maternal health care in post-conflict settings.

Timeliness: The study addresses a current and critical issue in global health, as the post-conflict period is a vulnerable time for healthcare systems. The findings are timely and relevant to ongoing efforts to rebuild and reform healthcare services in the DRC.

 Background

The introduction of the manuscript provides a generally informative background and adequately references relevant literature pertaining to maternal health care services and post-conflict settings. 

Research Design

The research design appears to be appropriate for the study's objectives. The manuscript outlines the use of nationally representative data to analyze maternal health care service utilization patterns in the post-conflict DRC. This approach allows for generalizability and provides a comprehensive view of the situation.

Methods

They adequately described the methods section, providing the information for readers to understand how the study was conducted. It includes details on data sources, sampling methods, and statistical analysis. The use of nationally representative data enhances the validity and reliability of the study's findings.

Results

The results section is generally clear and well-organized. It presents the findings in a structured manner with tables and figures to aid comprehension.

Conclusions:

The conclusions drawn in the manuscript appear to be supported by the results presented. The authors effectively link their findings to the broader context of maternal health care in post-conflict regions and discuss the implications for policy and practice.

Overall Assessment

The manuscript's introduction adequately provides background information and references relevant literature, with room for minor improvements in contextual detail and the inclusion of more recent references. The research design is appropriate, and the methods are adequately described. The results are clear. The conclusions are supported by the results.

I recommend acceptance of the manuscript for publication in its current form.

Reviewer 5 Report

Comments and Suggestions for Authors

This article assesses maternal health care services utilization in the post-conflict Democratic Republic of Congo and health inequalities over time. Authors assert that although improving maternal health is critical to achieving Sustainable Development Goals, the number of maternal deaths in sub-Saharan Africa attributed to a lack of access to and utilization of maternal health care services is still high.  Moreover, countries that have experienced armed conflict often have the most severe indicators of maternal mortality.  Authors contend that despite the fact that the Democratic Republic of the Congo (DRC) has many regions known for a series destabilizing conflicts and wars, a limited number of studies have utilized available datasets to assess the association between conflict and maternal healthcare services utilization in the DRC.  Therefore, authors aim to assess health inequality trends in selected maternal health care services utilization variables in post-conflict DRC using publicly available Demographic and Health Surveys.  The review is as follows:

1.       Within the abstract, write out acronyms (e.g., ANC, PNC)  in full  during their first introduction.

2.       Line 31 – write out the acronym WHO at its first introduction.

3.       Line 61 – Write out the acronym EDS-RDC at its first introduction.

4.       A brief description of the demographic profile of the DRC and related historical and sociocultural factors may help contextualize the inequalities of the ethnic groups.  Or, if space does not permit for the description of the demographic profile, an explanation of the urban and rural areas and major and minor ethnic groups in the DRC can be presented earlier in the paper.

Overall, this is a comprehensive, insightful paper on a unique topic. It can make an important contribution to the existing literature.  Addressing to some items can help make the paper clearer and improved.
